# Junctional conductance of retinal AII amacrine cell electrical synapses is decreased by NMDA receptors

Chloe Cable , Sidney P. Kuo and Eric A. Newman

*Department of Neuroscience, University of Minnesota, Minneapolis, Minnesota, USA*

Handling Editors: Nathan Schoppa & Conny Kopp-Scheinpflug

The peer review history is available in the Supporting Information section of this article (https://doi.org/10.1113/JP286537#support-information-section).

**Abstract figure legend** AII amacrine cells are an important class of interneuron in the vertebrate retina. In addition to traditional chemical synapses these cells communicate with other neurons and with each other via electrical synapses. **(A)** Electrical synapse strength between AII amacrine cells was measured using dual whole-cell patch-clamp electrophysiology. In electrically coupled cells a voltage step in one elicits a current response in the other. **(B)** Bath application of NMDA reduced junctional conductance at AII electrical synapses and the addition of NMDAR coagonists D-serine or glycine reduced conductance further. **(C)** Experiments were repeated in genetic mouse models lacking serine racemase (SR) or inositol 1,4,5-trisphosphate receptor type 2 (IP3R2), both thought to facilitate Müller cell D-serine gliotransmission. NMDAR-mediated plasticity was unaffected, indicating that Müller cell–derived D-serine is not necessary for this plasticity. This work adds to mounting evidence that NMDA receptors mediate plasticity at electrical as well as chemical synapses.

**Chloe Cable**, My interest in learning about the brain was sparked by my undergraduate studies at Tulane University. During my senior year I studied rat behaviour in opioid use disorder, which left me curious about the maladaptive plasticity that enables disease phenotypes. This curiosity led to a lifelong interest in the basic mechanisms underlying synaptic plasticity, which was the focus of my thesis work at the University of Minnesota. In my future career I will strive to continually uncover basic synaptic plasticity mechanisms so that we may further understand maladaptive plasticity in neurological diseases.

**Abstract** Retinal AII amacrine cells are extensively coupled together by electrical synapses. Changes to the strength of these synapses affect how signals are routed through rod and cone retinal pathways during scotopic and photopic vision. Plasticity at these electrical synapses has not, to date, been characterized using electrophysiological approaches. We investigated the effects of NMDA receptor (NMDAR) activation on electrical coupling between AII cells using dual whole-cell patch-clamp electrophysiology in mouse retinal slices. NMDAR activation substantially decreased junctional conductance between AII cells. Relieving the $Mg^{2+}$ block of NMDARs through bath application of $Mg^{2+}$-free solution or by depolarizing AII cells to 0 mV reduced junctional conductance. Exogenous application of NMDA decreased conductance between cells, a decrease which was blocked by the non-selective NMDAR antagonist D-APV but not by Ro 25-6981, a selective GluN2B-NMDAR antagonist. Addition of either D-serine or glycine, both NMDAR coagonists, without NMDA, reduced the junctional conductance and the addition of either coagonist to NMDA-treated retinas further decreased conductance. Experiments were conducted in inositol 1,4,5-trisphosphate receptor type 2 (IP3R2) knockout (KO) mice, serine racemase KO mice, and in wild-type (WT) mice with D-amino acid oxidase to reduce retinal D-serine levels. Under these conditions, the NMDAR-mediated decrease in conductance was maintained, indicating that endogenous D-serine is not necessary for NMDAR-mediated plasticity. These results demonstrate that NMDAR activation decreases electrical coupling between AII amacrine cells and suggest that both D-serine and glycine can serve as NMDAR coagonists for this plasticity.

(Received 11 March 2024; accepted after revision 12 December 2025; first published online 8 January 2026)

**Corresponding author** E. A. Newman: Department of Neuroscience, University of Minnesota, Minneapolis, MN 55455, USA. Email: ean@umn.edu

## Key points

- Retinal AII amacrine cells are extensively coupled together by electrical synapses.
- We show that NMDAR activation substantially decreased junctional conductance between AII cells.
- Relieving the $Mg^{2+}$ block of NMDARs reduced junctional conductance.
- Addition of either D-serine or glycine, both NMDAR coagonists, reduced the junctional conductance.
- This research adds to existing evidence that NMDA receptors contribute to the plasticity of a key electrical synapse in the retina, the electrical synapse coupling AII amacrine cells together.
- The study adds to mounting evidence that NMDARs mediate plasticity at electrical as well as chemical synapses.

## Introduction

Electrical synapses in the CNS facilitate fast, synchronous communication, bidirectional transmission of information, coincidence detection of subthreshold potentials, signal amplification and noise reduction through the flow of current across gap junctions (Nagy et al., 2018; Szczupak, 2016). Like chemical synapses, electrical synapses undergo plasticity, which manifests as increases or decreases in electrical coupling between neurons (Shimizu & Stopfer, 2013). Although electrical synapses are expressed throughout the brain, most research has focused on sensorimotor systems such as the retina, which expresses an abundance of electrical synapses that facilitate the processing of visual information. Previous studies have demonstrated plasticity of electrical synapses in the retina through neuromodulators such as adenosine and dopamine (Hampson et al., 1992; Li et al., 2013), the levels of which vary with background illumination and circadian rhythms (Godley & Wurtman, 1988; Ribelayga & Mangel, 2005; Wirz-Justice et al., 1984).

Many studies have focused on electrical synapses of the retinal AII amacrine cell, a well-characterized subclass of interneuron that plays a role in transmission of signals in both rod and cone pathways through chemical and electrical synapses (Famiglietti & Kolb, 1975; Graydon et al., 2018; Strettoi et al., 1992). Tracer

coupling experiments characterizing the effect of background illumination have implicated AII amacrine cell electrical synapses in serving as a 'light switch' between the rod and cone pathways (Demb & Singer, 2012). In dim light, electrically coupled AII cells facilitate the transfer of primary rod pathway signals through a cone pathway intermediary using electrical synapses between AII and ON cone bipolar cells (Demb & Singer, 2012). In bright light, electrical synapses between AII cells uncouple, preventing attenuation of cone-mediated signals through leakage into the AII cell network (Bloomfield & Völgyi, 2004). Additional tracer coupling experiments have suggested that both dopamine and NMDA receptors (NMDARs) modulate AII cell coupling (Hampson et al., 1992; Kothmann et al., 2012).

To date, all studies documenting neurotransmitter-induced plasticity in the AII electrical synapse network have utilized the tracer coupling technique, as detected by immunohistochemistry. Modulation of AII electrical synapses has not been characterized using dual whole-cell patch-clamp electrophysiology, which provides a direct measure of the strength of electrical coupling in real time. Although tracer coupling experiments yield a spatial representation of an electrically coupled network, dual patch-clamp recordings take advantage of directly measuring the strength of junctional conductance between two electrically coupled cells.

Here we report the effect of NMDAR activation on AII electrical synapse plasticity using dual whole-cell patch-clamp electrophysiology. We find that NMDAR activation decreases junctional conductance and that the addition of the NMDAR coagonists D-serine or glycine potentiates this NMDAR-mediated decrease in conductance.

## Materials and methods

### Ethical approval

All experimental procedures were approved by and adhered to the guidelines of the Institutional Animal Care and Use Committee of the University of Minnesota (IACUC number: D16-00288; protocol number: 1902-36831A). The authors understood and conformed to the ethical ARRIVE 2.0 guidelines for animal use in research.

### Animals

Retinal slices were prepared from C57BL/6J mice, which were housed in the University of Minnesota animal care facility in up to four animals per cage under a 12:12 h light/dark photocycle with *ad libitum* access to food and water. Experiments were conducted on 98 wild-type (WT; 52 male, 46 female; Jackson Lab #000664), 6 inositol 1,4,5-trisphosphate receptor type 2 (IP3R2) knockout (KO) (1 male, 5 female; obtained from Dr. J. Chen, University California, San Diego, La Jolla, CA, USA) and 7 serine racemase KO (SRKO; 3 male, 4 female; gift originally from Joseph Coyle; Basu et al., 2009) mice between the ages of 5 and 10 weeks.

### Retinal slice preparation

Experiments were conducted on 200 μm-thick retinal slices. Light-adapted mice were anaesthetized by inhalation of 4% isoflurane and killed by cervical dislocation. Eyes were removed and submerged in oxygenated (95% $O_2$, 5% $CO_2$) aCSF. After removal of the cornea, lens and vitreous, eyecups were hemisected and the retina was gently removed. Isolated retinae were embedded in low-temperature-gelling 4% agarose (Sigma-Aldrich, A0701) dissolved in aCSF and sliced in oxygenated aCSF using a vibratome. Individual slices were placed in a perfusion chamber, held in place with a platinum harp and perfused with oxygenated room temperature aCSF at 2.5 ml/min. Anaesthesia, dissection and slice preparation were performed under normal room illumination. During dual whole-cell patch-clamp electrophysiology and Neurobiotin tracer coupling experiments, lights were dimmed moderately and retinal slices were considered light adapted (Veruki & Hartveit, 2002).

### Identification of AII amacrine cells

Slices were visualized using infrared differential interference contrast microscopy and an Olympus FV1000 confocal microscope. AII amacrine cells were identified by the position of pear-shaped somata at the border between the inner nuclear and inner plexiform layers. Figure 1*A* shows an example of a pair of AII cells that were patched and dye-filled with the gap junction-impermeable Alexa Fluor 488 hydrazide (Sigma-Aldrich, A10436). AII amacrine cells have a characteristic primary stalk with proximal lobular appendages that form chemical synapses onto OFF cone bipolar and OFF ganglion cells. Distal, arboreal appendages that reach the ganglion cell layer contain electrical synapses that contact other AII amacrine cells and ON cone bipolar cells, as well as chemical synapses from rod bipolar cells (Strettoi et al., 1992).

### Electrical coupling measurement

Electrical coupling was assessed by dual whole-cell patch-clamp measurements of junctional conductance using a MultiClamp 700a amplifier (Molecular Devices)

and MATLAB-based, open-source Symphony software (https://symphony-das.github.io/). Borosilicate micropipettes (5–8 MΩ) filled with a $K^+$-based internal solution were used to patch onto pairs of AII cells within 30 μm of each other. Both cells of the pair were voltage clamped to −60 mV. Applying depolarizing steps to one cell resulted in characteristic outward currents in the depolarized cell and inward currents in the coupled cell (Fig. 1B; Veruki et al., 2003), confirming that the two cells were electrically coupled.

Data were collected by applying a family of voltage step pulses between −80 and 20 mV in 10 mV increments to one of the pair of voltage-clamped cells (Fig. 1B), followed by a −20 mV step applied to both cells simultaneously to monitor series resistance. Each voltage step consisted of a 500 ms baseline at −60 mV, a 500 ms voltage step and a 500 ms post stimulus baseline at −60 mV. Voltage-clamp currents were sampled at 20 kHz and filtered at 3 kHz. The 500 ms voltage steps were presented at 1.5 s intervals, and complete families of steps were repeated every 20 s. In preliminary experiments recordings were collected by applying the family of voltage steps to just one of the pair of cells throughout the entire experiment. In later experiments recordings were collected by alternately applying a family of voltage steps to each cell of the pair. The two approaches yielded similar values of calculated junctional conductance, and all data were combined.

## Analysis

The effect of drug treatment on electrical synapse strength was evaluated using dual whole-cell patch-clamp

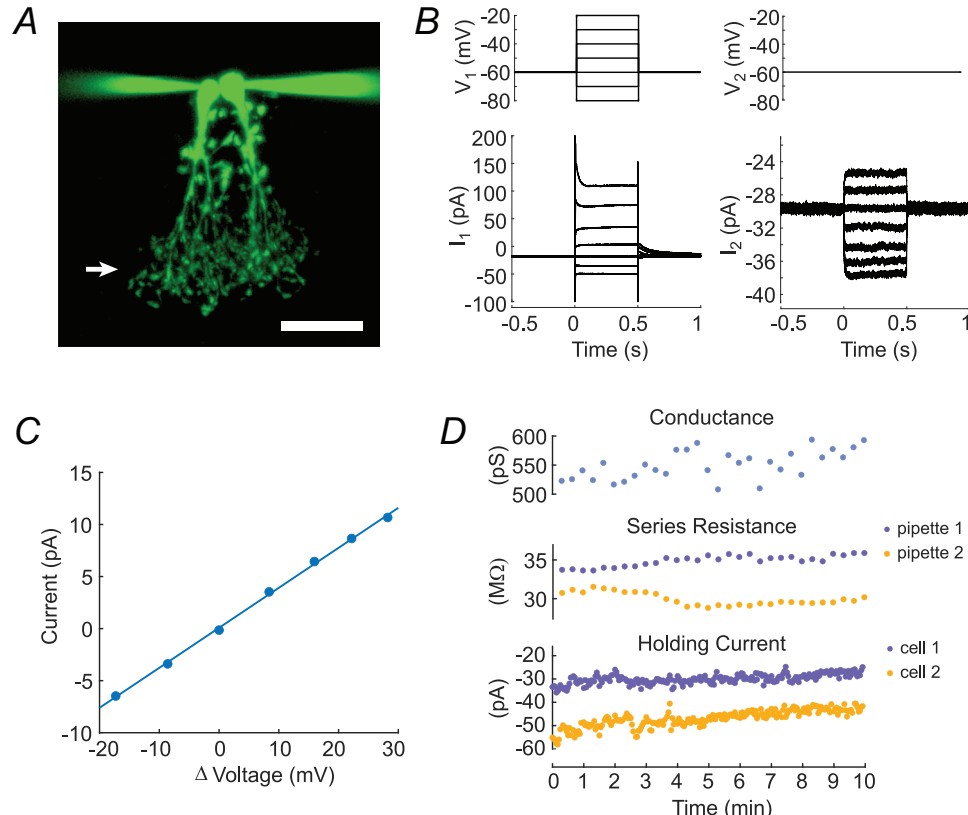

**Figure 1. Dual whole-cell patch-clamp measurement of retinal AII amacrine cell electrical synapse strength**

*A*, confocal image of a pair of coupled AII amacrine cells, dye-filled with the gap junction-impermeable Alexa Fluor 488 hydrazide. The two patch pipettes are seen to the left and right of the cell somata. The arrow indicates the location of homologous AII electrical synapses. For display purposes, the image was enhanced by adjusting the luminance factor gamma. Scale bar, 20 μm. *B*, traces (averages of 30 trials) showing measurement of junctional conductance. Cell 1 is given a family of voltage steps ($V_1$) from a holding potential of −60 mV, whereas Cell 2 remains voltage-clamped at −60 mV ($V_2$). Both cells respond to $V_1$ with changes in current ($I_1$ and $I_2$). *C*, for each voltage step the change in current $I_2$ is plotted against the difference between $V_1$ and $V_2$. Corrections are made for series resistance errors (see Materials and Methods for details). The junctional conductance is calculated as the slope of the least squares fit for each family of voltage steps. *D*, junctional conductance is monitored during the course of an experiment at 20 s intervals. Series resistance and holding current recorded from each pipette and cell are also monitored to track the integrity of the recordings (see Materials and Methods for details).

measurements of conductance during the time period when the drug was administered. Conductance was calculated (Fig. 1*C*) and plotted for the duration of an experiment (Fig. 1*D*). The health and stability of electrical recordings were monitored using holding current and series resistance measurements (Fig. 1*D*). Junctional conductance ($g_j$) of the electrical synapses between pairs of AII amacrine cells can be calculated as the current change in Cell 2 divided by the voltage difference between Cell 1 and Cell 2 following a voltage step in Cell 1, using equation (1) (Hartveit & Veruki, 2010),

$$g_j = \frac{I_j}{V_j} = \frac{I_2}{V_1 - V_2} \tag{1}$$

where $g_j$ is junctional conductance, $I_j$ is junctional current, $V_j$ is junctional voltage, $I_2$ is current in Cell 2, $V_1$ is voltage in Cell 1 and $V_2$ is voltage in Cell 2.

The series resistance of the pipettes used in recording from cells 1 and 2 can cause discrepancies between the voltage of the command step and the resulting voltage step induced in the cells. This discrepancy was compensated for by calculating the series resistance of each pipette and using equation (2) (Hartveit & Veruki, 2010),

$$g_j = \frac{I_j}{V_j} = \frac{-I_b + ((V_b - I_b \times R_{s2})/R_{m2})}{(V_a - I_a \times R_{s1}) - (V_b - I_b \times R_{s2})} \tag{2}$$

where $I_b$ is the current recorded in Cell 2, $V_b$ is the voltage command given to Cell 2, $R_{s2}$ is the series resistance of Cell 2 and Pipette 2, $R_{m2}$ is the membrane resistance of Cell 2, $V_a$ is the voltage command given to Cell 1, $I_a$ is the current recorded in Cell 1 and $R_{s1}$ is the series resistance of Cell 1 and Pipette 1. $R_s$ was estimated using Ohm's law by measuring the peak amplitude of the transient current response to a $-20$ mV voltage step commanded to cells 1 and 2 simultaneously. $R_m$ was calculated by subtracting $R_s$ from the total resistance as defined by the steady-state current response to the $-20$ mV step.

### Tracer coupling

Tracer coupling was assessed by patching onto an AII amacrine cell in a retinal slice and dialyzing the cell for 30 min with Neurobiotin. Borosilicate micropipettes (5–8 MΩ) contained a $K^+$-based internal solution using Alexa Fluor 488 hydrazide and 0.4% Neurobiotin Tracer (Vector labs, #SP-1120-50). Retinal slices were perfused with control or drug solution for 10 min prior to cell dialysis. After dialysis the pipette was carefully removed from the patched cell, and tracer was allowed to diffuse into coupled cells for an additional 60 min. Retinal slices were then fixed in 4% paraformaldehyde in 0.1 M phosphate-buffered saline (PBS) for 15 min and washed in PBS five times. Slices were then incubated in blocking solution containing 5% normal goat serum and 0.5%

Triton X-100 in PBS for 1 h and then incubated overnight at 4°C in a solution containing 1% normal goat serum, 0.5% Triton X-100 and 1:200 Streptavidin Alexa Fluor 568 conjugate (Invitrogen, S11226) in PBS. The next day slices were washed in PBS five times, mounted and imaged using an Olympus FV1000 confocal microscope and a 40× oil immersion objective.

Coupled AII amacrine cells were identified based on the shape and position of their somata within the inner nuclear layer. The effect of drugs on AII tracer coupling was analysed by quantifying the number of labelled AII amacrine cells (not including the dialyzed cell). Labelled ON cone bipolar cells were also quantified. Two blinded scorers counted the number of labelled cells, and the results were averaged. Cells were identified as labelled if their somata were brighter than background, as judged by the scorers.

### Solutions and drugs

Retinas were dissected, incubated, sliced and perfused in room temperature aCSF containing (in mM): 125 NaCl, 26 NaHCO$_3$, 1.25 NaH$_2$PO$_4$, 2.5 KCl, 1.5 MgCl$_2$, 20 glucose, 1.2 CaCl$_2$, 0.5 L-glutamine, 0.4 sodium ascorbate, and bubbled with 95% O$_2$/5% CO$_2$ (pH 7.35; 305–310 mOsm). For experiments performed in Mg$^{2+}$-free aCSF, MgCl$_2$ was omitted from the aCSF while maintaining the same pH and osmolality as aCSF containing Mg$^{2+}$. The patch pipette solution contained (in mM): 120 K-gluconate, 4.5 MgCl$_2$, 9 Hepes, 0.1 EGTA, 14 tris$_2$-phosphocreatine, 4 Na$_2$-ATP, 0.3 tris-GTP, as well as sucrose to bring the solution to 280–290 mOsm, pH 7.25. In preliminary electrical coupling experiments, as well as all tracer coupling experiments, the patch pipette solution contained 0.1% (w/v) Alexa Fluor 488 hydrazide to visualize AII amacrine cells. All drugs were applied by adding them to the bath solution.

Experimental drugs were purchased from the following suppliers: Sigma Aldrich: meclofenamic acid (MFA), gabazine, strychnine hydrochloride, D-serine, D-amino acid oxidase (DAAO), glycine. Tocris: 6-cyano-7-nitroquinoxalane-2,3-dione disodium salt (CNQX), NMDA, Ro 25–6981 maleate, D-(-)-2-amino-5-phosphonopentanoic acid (D-APV). Alomone Labs: tetrodotoxin citrate (TTX).

### Statistics

For statistical analysis of junctional conductance measurements, Student's paired *t* tests, one-way repeated-measures ANOVAs and two-way ANOVAs were used, as appropriate. For tracer coupling experiments, one-way ANOVAs were used in conjunction with Tukey-Kramer multiple comparisons *post hoc* tests

to make direct comparisons between conditions. All *P*-values are shown in the figures and are considered significant when $P < 0.05$. The statistical tests used in each experiment are described in the figure legends. Data are expressed as mean ± SD in the text and figures. In experiments measuring junctional conductance, sample size (*n*) represents the number of paired cell recordings for each group; in tracer coupling experiments *n* represents the number of dialyzed cells. Sample sizes are reported throughout the text. No more than one sample was collected from each slice, but multiple samples were collected from each mouse. Data analysis and statistics were carried out using custom MATLAB scripts. The data, images and MATLAB scripts for analysis and statistics will be made available upon request.

## Results

### Dual whole-cell patch-clamp measurement of retinal AII amacrine cell electrical synapse strength

We measured the junctional conductance between pairs of AII amacrine cells in mouse retinal slices using dual whole-cell patch-clamp electrophysiology. Previous studies have demonstrated that AII cells are electrically coupled to one another via distal, arboreal appendages in the proximal inner plexiform layer (Strettoi et al., 1992; Veruki & Hartveit, 2002). An example of a pair of dye-filled, electrically coupled AII amacrine cells is shown in Fig. 1*A*, with an arrow indicating the location of the homologous AII electrical synapses.

Previous studies have reported that junctional conductance measured with dual whole-cell patch-clamp drifts upward during recording. This drift is likely due to intracellular washout and changes in second messenger signalling (Veruki et al., 2008). We observed a similar

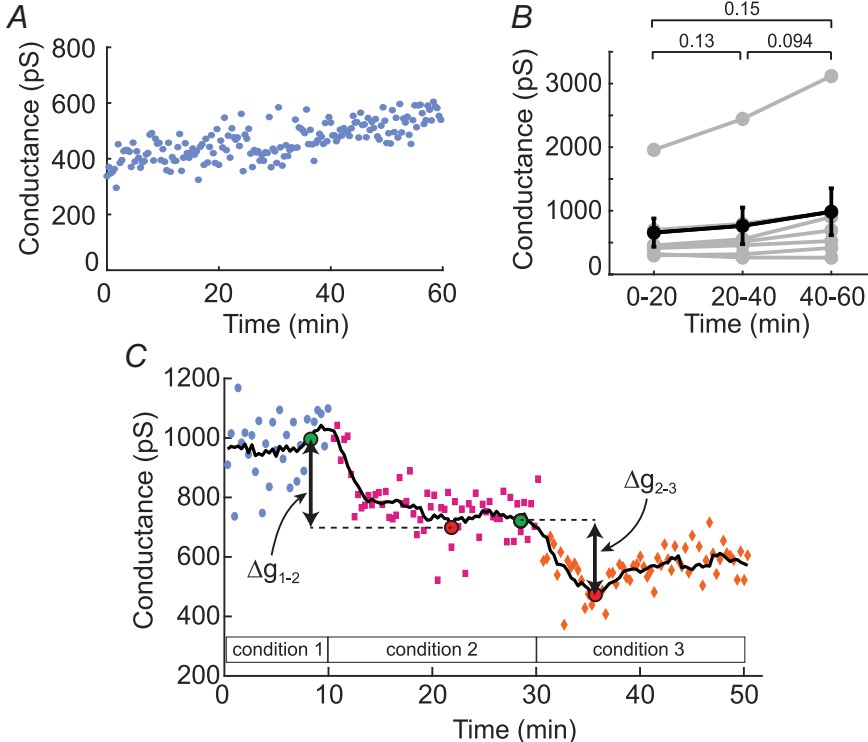

**Figure 2. All amacrine cell junctional conductance drifts upwards during experiments**
*A*, junctional conductance measurements from a single experiment in control aCSF. *B*, summary data showing the slow upward drift of conductance over time. Conductance was recorded for 60 min and averaged within 20 min time windows for comparison. Individual experiments (grey lines) and mean ± SD (black lines; *n* = 7 paired recordings from 5 mice). In this and subsequent figures, significance values (*P*) are shown in the figures. *C*, diagram depicting the calculation of conductance changes between experimental conditions, used to minimize the effects of conductance drift. A running average of all conductance measurements is first calculated (black line, 3 min time window). The change in conductance between two conditions caused by the addition of a pharmacological agent (e.g. $\Delta g_{1-2}$ between conditions 1 and 2) is then calculated as the conductance difference between the average of all measurements during the last 3 min before the end of condition 1 (first green circle) and the average of all measurements during the 3 min period bracketing the minimum peak in conductance during condition 2 (first red circle). The conductance change between condition 3 and condition 1 is calculated as $\Delta g_{1-2} + \Delta g_{2-3}$.

phenomenon in our recordings, with the degree of drift varying between experiments (Fig. 2*A*, *B*). We adopted the following procedure to minimize the effect of this drift when calculating changes in the junctional conductance in response to pharmacological manipulations (Fig. 2*C*). As the actual time course of the drift could not be directly measured during an experiment, this procedure does not fully correct for the drift. For each experimental session, a running average of all conductance measurements was calculated using a sliding window of 3 min width (Fig. 2*C*, black line). The running average was used to identify the maximal change in conductance due to a pharmacological manipulation. The effect of a given manipulation was calculated as the difference between the average of all measurements during the last 3 min before the manipulation (Fig. 2*C*, green circles) and the average of all measurements during the 3 min period bracketing the peak change in conductance, identified by the running average (Fig. 2*C*, red circles). For example the change in conductance between condition 1 and condition 2 in Fig. 2*C* was the difference between the conductances indicated by the first green circle and the first red circle in Fig. 2*C* ($\Delta g_{1-2}$). The change in conductance between condition 1 and condition 3 in Fig. 2*C* was estimated as $\Delta g_{1-2} + \Delta g_{2-3}$. All junctional conductance values and percentage changes in conductance reported in this work were calculated using this method. It was not possible to correct for drift by extrapolating the drift from the control baseline as the rate of drift sometimes increased during experiments, for example Fig. 6*A*. Because the drift (an increase in junctional conductance with time) was in the opposite direction to the pharmacologically induced changes in conductance we observed (a decrease in junctional conductance), the estimates of pharmacologically induced conductance changes using this correction method possibly underestimated the actual changes.

We validated our approach of measuring electrical synapse conductance with dual whole-cell patch-clamp using two pharmacological manipulations. We first evaluated the effect of MFA on junctional conductance. MFA blocks gap junctions and therefore should eliminate electrical synapse conductance. Indeed, the addition of 100 µM MFA reduced the junctional conductance to 0.6% of control, with the mean conductance dropping from $904 \pm 271$ pS (mean $\pm$ SD) in control solution to $5.20 \pm 8.69$ pS in MFA ($n = 5$ paired recordings from 3 mice; Fig. 3*A*, *B*).

Additionally, we evaluated the effect of blocking chemical synapses on junctional conductance by adding a cocktail of drugs: 10 µM CNQX to block AMPA receptors, 10 µM gabazine to block $GABA_A$ receptors, 1 µM strychnine to block glycine receptors and 0.3 µM TTX to block voltage-gated $Na^+$ channels (Fig. 3*C*–*F*). The drug cocktail greatly reduced chemical synaptic currents

recorded from the AII cells, as observed in the example traces showing the synaptic currents recorded from one cell of a pair (Fig. 3*C, D*), without altering the junctional conductance between the two cells of the pair (Fig. 3*E, F*). The synaptic currents, quantified by measuring the root mean square of the voltage-clamp currents, were reduced $83.6\% \pm 9.1\%$ by the drug cocktail. All remaining experiments reported here were conducted in the presence of this chemical synapse antagonist cocktail to isolate the effect of our manipulations to electrical synapses.

## NMDA receptor activation decreases AII electrical synapse conductance

NMDARs are commonly studied in the context of chemical synapse plasticity. Tracer coupling experiments in AII amacrine cells have implicated NMDARs as a potential modulator of electrical synapses as well (Kothmann et al., 2012). We hypothesized that the activation of NMDARs would modulate AII electrical synapse junctional conductance.

Activation of NMDARs requires the release of the $Mg^{2+}$ block inherent to NMDARs. Bath application of $Mg^{2+}$-free aCSF relieves the $Mg^{2+}$ block and allows for the activation of NMDARs (Nowak et al., 1984). We observed a decrease in junctional conductance between AII cells when switching from $Mg^{2+}$-containing solution to Mg-free solution. Junctional conductance was reduced to 57.7% of control, from $442 \pm 97.0$ pS in $Mg^{2+}$-containing solution to $255 \pm 151$ pS in $Mg^{2+}$-free solution ($n = 5$ paired recordings from 2 mice; Fig. 4*A*, *B*). This result suggests that NMDAR activation may play a role in decreasing junctional conductance at these synapses.

A second approach to relieving the $Mg^{2+}$ block of NMDARs is to depolarize the cell. The holding potential under voltage clamp was raised from $-60$ to $0$ mV in both cells of recorded pairs, and the same family of voltage steps relative to the holding potential was administered. The junctional conductance between AII cells decreased to 41.4% of control, from $526 \pm 178$ pS at $-60$ mV to $218 \pm 118$ pS at $0$ mV ($n = 5$ paired recordings from 4 mice; Figs. 4*C*, *D*). Taken together these results indicate that NMDARs may contribute to the regulation of AII electrical synapses. Note that both the $Mg^{2+}$-free and depolarization experiments were performed without the addition of NMDA or glutamate, suggesting that under our experimental conditions there was sufficient endogenous glutamate in the retinal extracellular space to activate NMDARs to some extent when the $Mg^{2+}$ block was removed. From this point forward, all experiments investigating the role of NMDARs in modulating AII electrical synapses were performed in $Mg^{2+}$-free aCSF

containing the chemical synapse antagonist cocktail while clamping both cells at −60 mV holding potential.

We directly tested whether the activation of NMDARs modulates AII amacrine cell electrical coupling by addition of NMDA, as well as the NMDAR coagonist D-serine. A coagonist, either D-serine or glycine, is necessary for activation of NMDARs (Johnson & Ascher, 1987; Mothet et al., 2000). Addition of 100 μM NMDA and 200 μM D-serine led to a decrease in junctional conductance to 46.9% of control, with conductance decreasing from 508 ± 181 pS to 238 ± 127 pS (n = 6 paired recordings from 3 mice; Fig. 4E, F). Experiments were also conducted in the presence of the non-selective NMDAR antagonist D-APV to test whether NMDA was acting specifically on NMDARs. In the presence of 50 μM D-APV, junctional conductance did not change after the addition of NMDA + D-serine, equalling 734 ± 417 pS in control solution and 769 ± 501 pS in NMDA + D-serine (n = 6 paired recordings from 5 mice; Fig. 4G, H).

Two NMDAR subtypes are commonly implicated in chemical synapse plasticity: GluN2A and GluN2B subunit-containing NMDARs. GluN2B-containing receptors have been shown to colocalize with Cx36 gap junctional proteins in AII amacrine cell arboreal dendrites (Veruki et al., 2019). We tested whether NMDA-GluN2B receptors mediate electrical synapse modulation by applying the GluN2B-selective antagonist Ro 25–6981 prior to the addition of NMDA and D-serine. We found, however, that addition of Ro 25–6981 did not block the NMDA + D-serine-mediated decrease in conductance. In the presence of 3 μM Ro 25–6981, NMDA + D-serine reduced junctional conductance to 44.3% of control, from 469 ± 126 pS to 208 ± 136 pS (n = 5 paired recordings from 5 mice; Fig. 4I, J). The similar reduction in gap junctional conductance by NMDA and D-serine coapplication in both control conditions and in the presence of Ro 25–6981 suggests that GluN2B subunit-containing NMDARs do not mediate

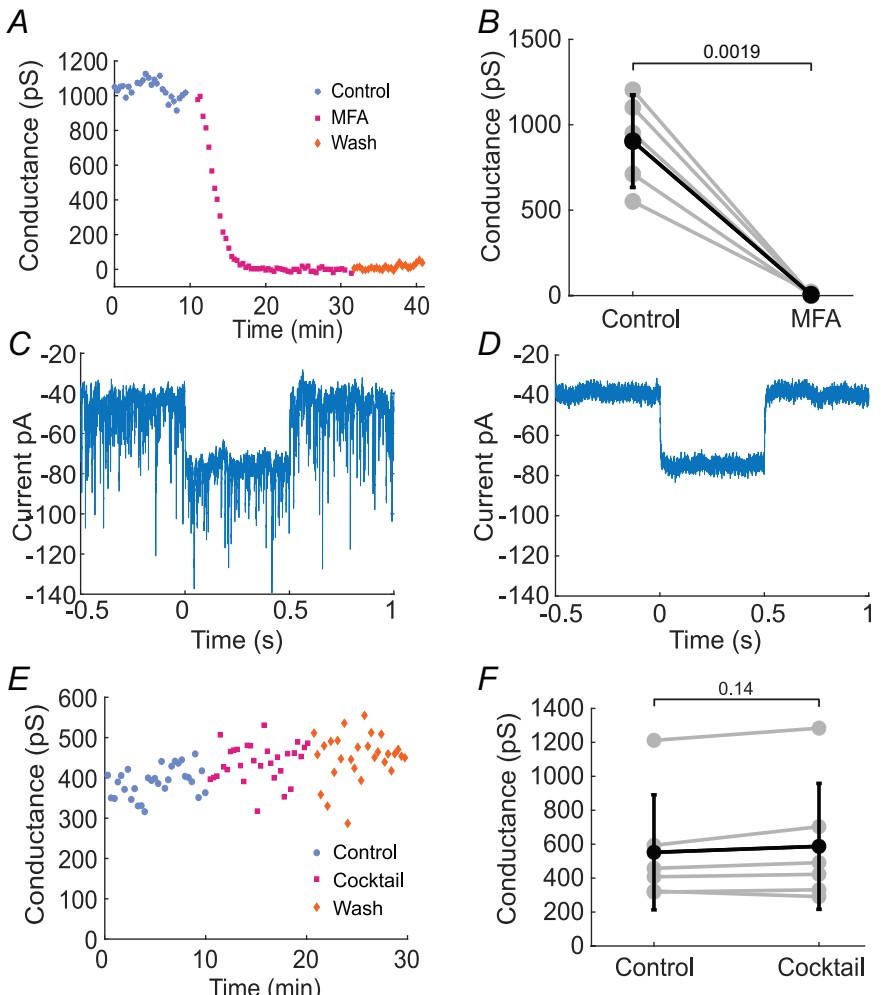

**Figure 3. Validation of AII amacrine cell junctional conductance measurements**
*A and B*, AII amacrine cell junctional conductance is eliminated by addition of the gap junction blocker MFA. *A*, example experiment and summary data (*B*) showing that MFA reduces the junctional conductance between AII cells (n = 5 paired recordings from 3 mice). *C–F*, electrical coupling between AII cells is preserved following block of chemical synaptic transmission. Traces of currents (single trials) recorded from Cell 2, responding to a 40 mV depolarizing voltage step in Cell 1, in the absence (*C*) and in the presence (*D*) of chemical synapse antagonists and channel blocker CNQX, gabazine, strychnine and TTX, blocking AMPA, GABA_A and glycine receptors and voltage-gated Na+ channels, respectively. The antagonist cocktail silenced the robust chemical synaptic inputs to AII amacrine cells. All data depicted hereafter were collected in the presence of the antagonist cocktail. *E*, example experiment and summary data (*F*) showing that blocking chemical synaptic transmission does not alter electrical coupling between AII cells (n = 6 paired recordings from 3 mice). *B* and *F*, individual experiments (grey lines) and mean ± SD (black lines). *P*-values are calculated using Student's paired *t* test.

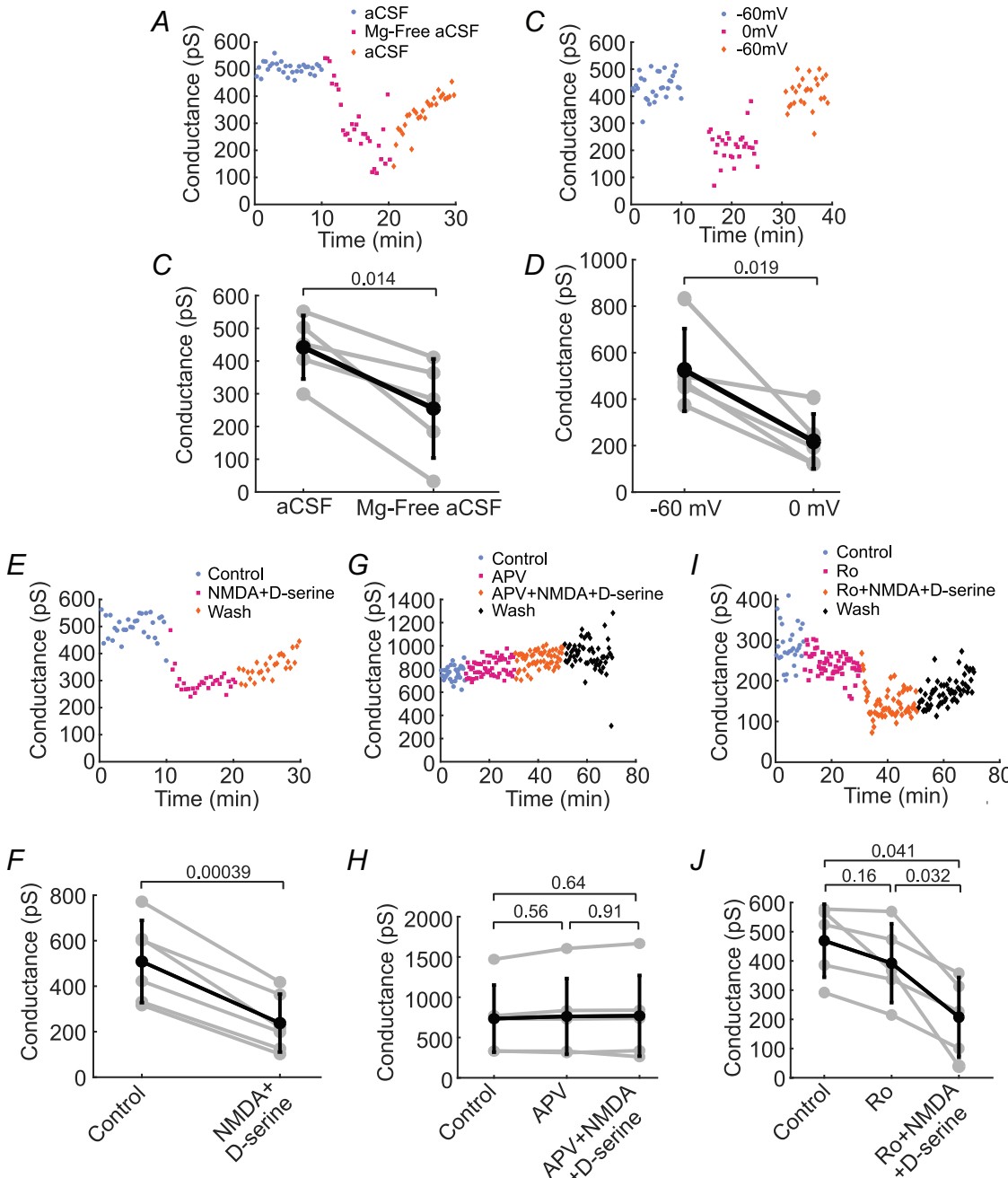

**Figure 4. NMDA receptor activation decreases AII cell junctional conductance**

*A–D*, AII cell junctional conductance is decreased by relief of the NMDAR $Mg^{2+}$ block. *A*, example experiment and summary data (*B*) showing that removal of $Mg^{2+}$ from the aCSF results in a decrease in junctional conductance between AII cells ($n = 5$ paired recordings from 2 mice). *C*, example experiment and summary data (*D*) showing that membrane potential depolarization decreases junctional conductance between AII cells ($n = 5$ paired recordings from 4 mice). All data depicted hereafter were collected in $Mg^{2+}$-free aCSF. *E*, example experiment and summary data *(F)* showing that NMDAR activation by the addition of NMDA and the coagonist D-serine decreases conductance between AII cells ($n = 6$ paired recordings from 3 mice). *G*, example experiment and summary data (*H*) demonstrating that block of NMDARs with D-APV prevents NMDAR-mediated decreases in junctional conductance between AII cells ($n = 6$ paired recordings from 5 mice). *I*, example experiment and summary data (*J*) demonstrating that block of GluN2B-NMDARs with the selective antagonist Ro 26–6981 (Ro) does not prevent NMDAR-mediated decreases in junctional conductance between AII cells ($n = 5$ paired recordings from mice). *B, D, F, H, J*, Individual experiments (grey lines) and mean $\pm$ SD (black lines). *P*-values in (*B*), *(D)*, *(F)* calculated using paired *t* test. *P*-values in (*H*) and (*J*), calculated using one-way repeated-measures ANOVA followed by Tukey-Kramer *post hoc* multiple comparisons.

the observed NMDAR-evoked decrease in junctional conductance.

## NMDA receptor activation decreases tracer coupling between AII amacrine cells

A previous study using Neurobiotin tracer to assess coupling between AII amacrine cells suggests that NMDAR activity increases electrical synapse strength (Kothmann et al., 2012). Because our junctional conductance measurements indicate the opposite effect of NMDAR activation, we tested the effect of both NMDAR agonism and antagonism on Neurobiotin tracer coupling in our retinal slice preparation (Fig. 5). We found that 100 μM NMDA + 200 μM D-serine significantly reduced the number of tracer-coupled AII cells compared to control, from $3.31 \pm 1.90$ to $1.10 \pm 1.14$ cells (Fig. 5C; n: control, 13; NMDA + D-serine, 15. The NMDAR antagonist APV (50 μM) increased the number of tracer-coupled cells compared to control (Fig. 5C, $5.04 \pm 2.82$, $n = 13$), but this difference was not significant. The NMDAR-mediated decrease in tracer-coupled AII cells mirrors the observed NMDAR-mediated decrease in junctional conductance seen in our dual whole-cell patch-clamp experiments.

Many bipolar cells were also labelled in the tracer coupling experiments (Fig. 5A, B) although no clear pattern of the number of coupled cells was seen among the three experimental conditions. The numbers of tracer-coupled bipolar cells resulting from AII amacrine cell tracer dialysis in the three conditions were as follows: control, $14.04 \pm 9.20$; NMDA + D-serine, $17.4 \pm 12.7$; APV, $10.7 \pm 10.7$ cells (no significant differences; control *vs.* NMDA, $P = 0.658$; control *vs.* APV, $P = 0.673$; n: control, 13; NMDA + D-serine, 15; APV, 13; one-way ANOVA followed by Tukey-Kramer multiple comparisons).

## NMDAR coagonist D-serine potentiates, but is not necessary, for NMDAR-mediated decreases in AII cell junctional conductance

It is well known that agonist binding alone to NMDARs is insufficient for channel opening and that the presence of a coagonist is necessary (Johnson & Ascher, 1987). Determining the coagonists involved in NMDAR activation may hint at the pathways or circuitry that induces NMDAR-mediated plasticity. We probed the role of the NMDAR coagonist D-serine in AII electrical synapse plasticity by evaluating the effect of application of D-serine alone. Addition of 200 μM D-serine yielded a small but significant decrease in junctional conductance to 85.6% of control, from $646 \pm 268$ pS to $533 \pm 299$ pS ($n = 5$ paired recordings from 3 mice; Fig. 6A, B). Next we determined the effect of D-serine on NMDA-mediated

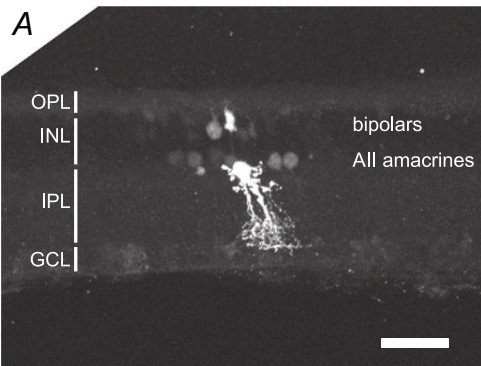

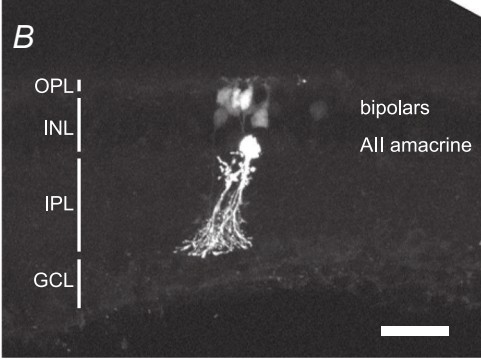

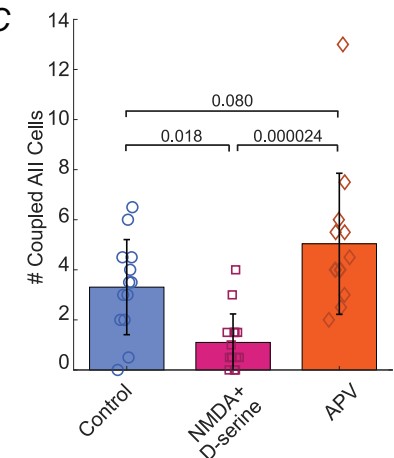

**Figure 5. Tracer coupling between AII amacrine cells**
*A, B,* confocal fluorescence micrographs of retinal slices showing a brightly labelled AII amacrine cell dialyzed with Neurobiotin and AII amacrine cells and bipolar cells coupled with the dialyzed cell. Images are Z-stacks of confocal sections showing all labelled cells in the retinal slice. Scale bars, 50 μm. OPL, outer plexiform layer; INL, inner nuclear layer; IPL, inner plexiform layer; GCL, ganglion cell layer. *A,* control solution. Several AII amacrine cells, as well as bipolar cells, are coupled with the dialyzed AII cell. *B,* NMDA + D-serine. No other AII amacrine cells are coupled with the dialyzed AII cell in this image. *C,* summary bar graph showing a NMDA + D-serine-mediated decrease in coupling between AII amacrine cells. *P*-values are calculated using one-way ANOVA followed by Tukey-Kramer *post hoc* multiple comparisons.

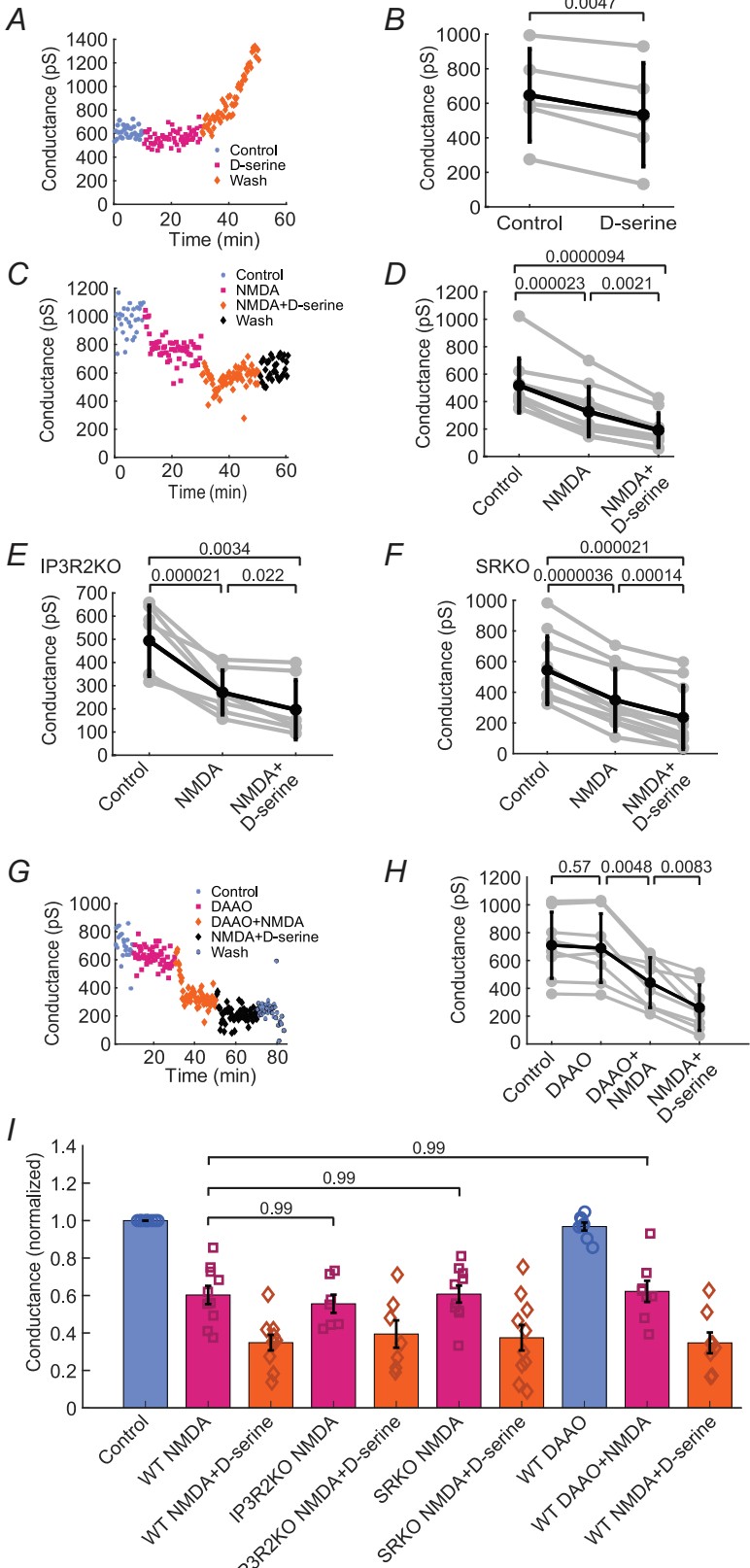

**Figure 6. The NMDAR coagonist D-serine potentiates but is not necessary for NMDAR-mediated decreases in AII cell junctional conductance**

*A*, example experiment and summary data (*B*) showing that D-serine alone decreases junctional conductance between AII cells (*n* = 5 paired recordings from 3 mice). *C*, example experiment and summary data (*D*) showing that NMDA alone decreases junctional conductance between AII cells and that addition of D-serine along with NMDA further decreases the conductance (*n* = 10 paired recordings from 7 mice). *E, F*, NMDA-mediated decreases in conductance are not prevented in genetic models meant to decrease D-serine levels. *E*, experiments on retinas from IP3R2KO mice, which have reduced Ca$^{2+}$ signalling in Müller glial cells. Summary data showing that the NMDA-mediated conductance decrease between AII cells is maintained in these mice (*n* = 7 paired recordings from 6 mice). *F*, experiments on retinas from SRKO mice, where production of D-serine is blocked. Summary data showing that the NMDA-mediated conductance decrease between AII cells is also maintained in these mice (*n* = 10 paired recordings from 8 mice). *G*, example experiment and summary data (*H*) demonstrating that the presence of DAAO, which degrades D-serine, does not prevent the NMDA-mediated decrease in conductance (*n* = 8 paired recordings from 7 mice). *I*, summary bar graph showing that the NMDA-mediated decreases in AII junctional conductance are maintained in IP3R2KO and SRKO mice, as well as in WT mice following DAAO addition. NMDA-mediated conductance decreases (purple bars) under these conditions are all similar to those observed in WT mice. All data are normalized to the conductance in control aCSF for each condition and show individual experimental data and the mean ± SD. *B, D–F, H*, individual experiments (grey lines) and mean ± SD (black lines). *B*, *P*-values are calculated using paired *t* test. *D-F, H*, *P*-values are calculated using one-way repeated-measures ANOVA followed by Tukey-Kramer *post hoc* multiple comparisons. *I*, *P*-values are calculated using two-way ANOVA followed by Tukey-Kramer *post hoc* multiple comparisons.

changes in conductance. To do this, we first applied 100 μM NMDA alone for 20 min, followed by 100 μM NMDA in conjunction with 200 μM D-serine for another 20 min. NMDA alone decreased conductance to 63.2% of control, from $517 \pm 197$ pS to $327 \pm 179$ pS ($n = 10$ paired recordings from 7 mice; Fig. 6*C, D*). The addition of D-serine along with NMDA further decreased the conductance, from $327 \pm 179$ pS in NMDA to $192 \pm 123$ pS in NMDA + D-serine ($n = 10$ paired recordings from 7 mice; Fig. 6*C, D*).

Because coagonist binding is required for NMDAR activation (Johnson & Ascher, 1987), and NMDA application alone led to a decrease in junctional conductance, D-serine or some other coagonist must already be present in the tissue. We employed IP3R2 and SRKO genetic mouse models, which are thought to reduce D-serine levels in the retina, to explore this possibility.

D-serine is a known gliotransmitter, and its release from glia is triggered by astrocytic $Ca^{2+}$ increases (Mothet et al., 2005) mediated by IP3R2 (Henneberger et al., 2010; Sherwood et al 2017). Barrel cortex stimulation of IP3R2 KO mice (IP3R2KO) yields decreased D-serine release compared to WT mice, likely due to decreased astrocytic $Ca^{2+}$ levels (Takata et al., 2011). Müller cells, the primary glial cells of the retina (Newman & Reichenbach, 1996), also display decreased glial $Ca^{2+}$ signalling in IP3R2KO (Biesecker et al., 2016) and, therefore, are likely to have decreased D-serine release. We hypothesized that compared to WT retinas, IP3R2KO retinas treated with NMDA alone would show a smaller junctional conductance decrease, and that only with the addition of exogenous D-serine would NMDA cause the same magnitude of decrease as seen in the WT retina. Contrary to this expectation, the NMDA-evoked decrease in conductance in IP3R2KO retinas did not differ from that seen in WT retinas. Junctional conductance decreased to 54.9% of control, from $494 \pm 152$ pS to $271 \pm 95.1$ pS ($n = 7$ paired recordings from 6 mice; Fig. 6*E*), indicating that either D-serine release in IP3R2KO tissue is similar to that in WT, or that glycine, the other known NMDAR coagonist, served as the coagonist.

We then repeated the previous experiments in serine racemace KO (SRKO) mice to determine if D-serine is necessary to mediate the decrease in NMDA-evoked conductance. Serine racemace is an enzyme that converts L-serine to D-serine and mice deficient in the enzyme lack D-serine in their retinas (Wolosker, Sheth et al., 1999; Sullivan et al., 2011). Similar to above, we hypothesized that compared to WT retinas the NMDA-evoked conductance decrease in SRKO retinas would be smaller, and that only with the addition of exogenous D-serine would the NMDA-induced decrease in conductance be as large as in WT mice. Again, as observed in the IP3R2KO mice, the NMDA-evoked conductance decrease in SRKO retinas did not differ from that in WT retinas, decreasing

to 64.2% of control, from $544 \pm 221$ pS to $349 \pm 202$ pS ($n = 10$ paired recordings from 8 mice; Fig. 6*F*).

It was possible that we observed negative results in the genetic models due to compensatory mechanisms in the global knockouts leading to increased D-serine levels. To confirm that D-serine is not necessary for NMDA-mediated decreases in junctional conductance we used another approach in WT mice. We first treated retinas with D-amino acid oxidase (DAAO), an enzyme which breaks down D-serine in the tissue. After a 20 min incubation in 300 μg/ml DAAO, NMDA + DAAO was added followed by NMDA + D-serine in the absence of DAAO. DAAO treatment did not prevent the NMDA-mediated decrease in conductance. Conductance equalled $710 \pm 238$ pS in control and $689 \pm 247$ pS in DAAO and decreased to 62.1% of control in NMDA + DAAO ($441 \pm 180$ pS, $n = 8$ paired recordings from 7 mice) and to 36.8% of control in NMDA + D-serine ($261 \pm 161$ pS) (Fig. 6*G, H*). These results indicate that D-serine can act as a coagonist for NMDAR-mediated decreases in conductance, but is not necessary for NMDAR-mediated plasticity.

A summary of the genetic model and DAAO results is given in Fig. 6*I* and shows that there are no significant differences in NMDA-mediated junctional conductance decreases in IP3R2KO and SRKO mice and in DAAO experiments, compared to WT.

## Glycine potentiates NMDAR-mediated decreases in conductance at AII electrical synapses

Glycine is the other known NMDAR coagonist besides D-serine (Johnson & Ascher, 1987; Mothet et al., 2000). Because none of our manipulations to lower D-serine in the retina reduced NMDA-mediated decreases in AII electrical coupling, we hypothesized that endogenous glycine could also serve as the coagonist for NMDA-mediated changes in coupling. To test this, we first added 400 μM glycine alone to the tissue. Similar to the results when D-serine was added alone, glycine yielded a small, but significant, decrease in junctional conductance, with conductance decreasing to 86.3% of control following glycine addition. Conductance decreased from $570 \pm 146$ pS to $492 \pm 99.1$ pS ($n = 6$ paired recordings from 3 mice; Fig. 7*A, B*). In the following experiment, NMDA and then NMDA + glycine were added sequentially. Addition of NMDA alone decreased the conductance to 48.1% of control, from $801 \pm 620$ pS to $385 \pm 357$ pS ($n = 6$ paired recordings from 4 mice; Fig. 7*C, D*). Addition of NMDA + glycine then led to a further decrease in conductance to 28.7% of control from $385 \pm 357$ pS in NMDA to $230 \pm 182$ pS in NMDA + glycine ($n = 6$ paired recordings from 4 mice; Fig. 7*C, D*). These results mirror the effects that were observed

with D-serine and indicate that both glycine and D-serine may play a role as coagonists in NMDAR-mediated AII electrical synapse plasticity.

## Discussion

A variety of methods have been used in previous studies to analyse electrical synapse strength, including tracer coupling, electrical synchronization, noise analysis, and coupling coefficients (Connors, 2017; Nagy et al., 2018). Here we have used dual whole-cell patch-clamp electrophysiology to assess electrical coupling in AII amacrine cells, a method that directly measures electrical synapse strength. We found that NMDAR activation substantially reduces junctional conductance and that the addition of the NMDAR coagonists D-serine or glycine further decreases conductance. NMDA modulation of coupling was not prevented by blocking NMDARs containing the GluN2B subunit, indicating that these receptors are not responsible for the observed electrical synapse plasticity. These results represent the first demonstration of neurotransmitter-mediated plasticity at AII amacrine cell electrical synapses as measured directly by an electrophysiological approach.

This study is just one of two that addresses NMDAR-mediated plasticity at AII electrical synapses.

Our finding that the activation of NMDARs decreases electrical coupling between AII amacrine cells is at odds with the previous study of Kothmann et al., who used the Neurobiotin tracer method to assess coupling between AII cells (Kothmann et al., 2012). Kothmann et al. reported that blockade of NMDA receptors reduced tracer coupling between AII cells, whereas we found that the activation of NMDARs decreased the junctional conductance between AII cells. To determine whether the conflicting results were due to the difference in techniques employed, we repeated our measurements using the Neurobiotin tracer method. The tracer results confirmed our junctional conductance results; NMDAR activation reduced tracer coupling as well as junctional conductance.

Several factors could account for the discrepancy between our results and those of Kothmann et al. Kothmann et al. used an NMDAR antagonist in their experiments, whereas we evaluated the effect of both NMDAR agonists and antagonists. Although both studies were done under light-adapted conditions, our experiments were performed in retinal slices using patch pipettes, whereas Kothmann et al. used retinal whole mounts and sharp electrodes. We used retinal slices and patch pipettes so that we could directly compare our junctional conductance and tracer coupling results.

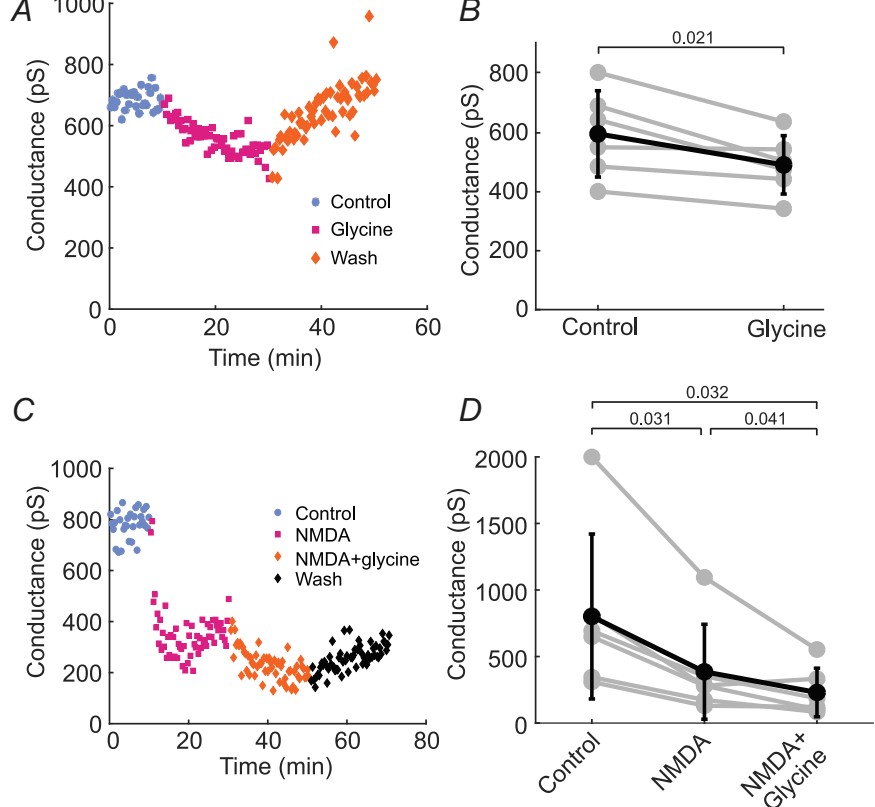

**Figure 7. The NMDAR coagonist glycine potentiates the NMDAR-mediated decrease in AII cell junctional conductance**

*A*, example experiment and summary data (*B*) showing that glycine addition alone decreases junctional conductance between AII cells (*n* = 6 paired recordings from 3 mice). *C*, example experiment and summary data (*D*) showing that NMDA alone decreases junctional conductance between AII cells, and that addition of glycine along with NMDA further decreases the conductance (*n* = 6 paired recordings from 4 mice). *B, D*, individual experiments (grey lines) and mean ± SD (black lines). *B*, *P*-values are calculated using paired *t* test. *D*, *P*-values are calculated using one-way repeated-measures ANOVA followed by Tukey-Kramer *post hoc* multiple comparisons.

Notably, our experiments were performed in mouse, whereas Kothmann et al. used rabbit. Previous studies have revealed differences in coupling between retinal neurons in different species. Although Tsukamoto et al. asserted that the canonical scotopic circuitry among mammals is conserved, they also found that some scotopic microcircuitry varied across species (Tsukamoto et al., 2001). Although electrical coupling between AIIs and ON cone bipolar cells remains conserved between mouse and rabbit, rabbit bipolar cells have significantly more gap junctional contacts compared to mouse. In addition, rabbit bipolar cells are extensively coupled with other bipolar cells of the same class, a pattern that is not observed in the mouse (Sigulinsky et al., 2020). Potential species differences in our study *vs.* that of Kothmann et al. may be clarified as studies continue to probe this topic.

Our study is the first to demonstrate direct electrophysiological evidence of AII–AII electrical synapse plasticity, as well as to observe congruent results using both electrophysiology and tracer coupling techniques. Our study neither mirrors nor contradicts that of Kothmann et al.; rather both studies add to our growing knowledge of AII–AII electrical synapse plasticity using various animal models and techniques.

It is important to note that inconsistent results across techniques used to evaluate electrical synapse plasticity between AII cells, as well as other neurons, are not uncommon. Using tracer coupling, several groups have published robust evidence showing that dopamine receptor activation decreases coupling within the AII amacrine cell network (Hampson et al., 1992; Kothmann et al., 2009; Urschel et al., 2006). Yet, others have reported that measuring junctional conductance directly with paired recordings does not support a role for dopamine in modulating electrical synapses between AII cells (Demb & Singer, 2012; Hartveit & Veruki, 2012). Contradictory results concerning the effects of NMDA on electrical coupling in the inferior olive have also been reported (Mathy et al., 2014; Turecek et al., 2014). These conflicting findings indicate that the modulation of electrical coupling may depend on multiple processes that could vary depending on the preparations and techniques used to assess coupling.

We have shown that both D-serine and glycine can serve as coagonists in NMDA-mediated modulation of coupling between AII amacrine cells, further reducing AII cell junctional conductance. Different release mechanisms for the two coagonists suggest that various pathways may be involved in this plasticity. Glycine is released from AII arboreal dendrites (Grimes et al., 2022) as well as other small-field amacrine cells (Menger et al., 1998), a subset of which are activated by the ON pathway and release glycine onto AII arboreal dendrites with illumination (Marc et al., 2014).

The source of D-serine is less clear, and there is an ongoing debate whether the coagonist is released from glial cells or neurons (Wolosker, Blackshaw et al., 1999; Miya et al., 2008). We tested whether the D-serine which potentiates NMDA-mediated electrical coupling plasticity originates from glial cells. D-serine is thought to be released from glia in a $Ca^{2+}$-dependent manner (Mothet et al., 2005). We found that the NMDA-mediated decrease in junctional conductance was not reduced in IP3R2KO mice, which show reduced $Ca^{2+}$ signalling in Müller cells (Biesecker et al., 2016), the glial cells present in the inner plexiform layer (Newman & Reichenbach, 1996). This result suggests that the D-serine that contributes to AII amacrine cell plasticity does not originate from retinal glial cells. However, D-serine release from Müller cells may not be solely dependent on IP3R2 receptor $Ca^{2+}$ increases. Sherwood et al. have shown that other glial IP3Rs besides IP3R2 mediate D-serine release (Sherwood et al., 2017), and Shigetomi et al. have shown that astrocytic TRPA1 $Ca^{2+}$ activity can mediate D-serine release and subsequent long-term potentiation at hippocampal synapses (Shigetomi et al., 2013). In addition, the presence of glycine may compensate for decreased D-serine release in IP3R2KO mice.

Although we demonstrated that exogenous NMDA application to retinal slices reduces junctional conductance between AII cells, the source of the endogenous glutamate that could enable this plasticity remains unclear. Scotopic and photopic lighting conditions differentially activate rod and cone bipolar cells, which dictates ambient glutamate tone in the IPL that may affect AII cell NMDAR activation. Rod bipolar cells form glutamatergic chemical synapses onto AII arboreal dendrites (Sterling et al., 1988), which contain extrasynaptic NMDARs localized near electrical synapses (Veruki et al., 2019). Veruki et al. (2019) demonstrated that the activation of bipolar cells leads to presynaptic glutamate transporter activity on nearby synapses through glutamate spillover (Veruki et al., 2006). Extrasynaptic AII NMDAR activation could occur by the same mechanism. However, several studies indicate that glutamate release at these synapses exclusively activates AMPARs, not NMDARs (Grimes et al., 2014; Singer & Diamond, 2003; Trexler et al., 2005). ON cone bipolar cells do not form direct glutamatergic synapses onto AII cells. However, they do synapse onto ganglion cells, and spillover at these synapses activates extrasynaptic NMDARs on ganglion cell dendrites (Chen & Diamond, 2002), which localize to the same sublamina of the IPL as AII electrical synapses and NMDARs. Kothmann et al. suggest that increased phosphorylation of AII Cx36 proteins and subsequent modulation of electrical synapses are a result of bipolar-ganglion cell synapse spillover (Kothmann et al., 2012), though no other studies have investigated this possibility.

Although we speculate on the mechanisms behind NMDAR activation, the location of the NMDARs that modulate AII junctional conductance in our study cannot be definitively localized to AII amacrine cells. The effects of our pharmacological manipulations could be due to activation of NMDARs on upstream neurons rather than on the AII cells themselves. AII amacrine cells receive input from as many as 28 distinct classes of neurons (Marc et al., 2014). We controlled for this by suppressing general chemical synapse communication through a cocktail of antagonists. However, this did not block all chemical synapse communication, nor the electrical synapse communication that occurs between AII amacrine and ON cone bipolar cells. Out of the five subtypes of murine ON cone bipolar cells (Ghosh et al., 2004; Wassle et al., 2009) four may share electrical synapses with AII amacrine cells (Petrides & Trexler, 2008). Using electron microscopy, Tsukamoto and Omi have suggested that all five subtypes of ON cone bipolar cells are electrically coupled to AII amacrine cells to a varied extent (Tsukamoto & Omi, 2017). It should be noted, however, that our dual whole-cell patch-clamp experiments directly measured the conductance of the electrical synapses between AII cells. In addition, our depolarization experiments, which resulted in reduced junctional conductance, directly depolarized the AII cells, suggesting that the NMDARs responsible for junctional conductance modulation are localized to AII cells. While rodent AII cells express NMDARs that colocalize with Cx36 gap junction proteins (Veruki et al., 2019), interventions that specifically target AII amacrine cell receptors would address the definitive location of NMDAR modulation.

Our finding that NMDAR activation decreases AII electrical synapse strength suggests a possible mechanism by which AII amacrine cells contribute differentially to the processing of both rod and cone signals in the retina. Tracer coupling and receptive field sizes of AII cells are maximal at scotopic light levels (Bloomfield et al., 1997; Bloomfield & Völgyi, 2004). Extensive electrical coupling within the AII cell network under these conditions is proposed to enhance the sensitivity of rod-mediated vision by reducing noise within the rod bipolar cell pathway (Dunn et al., 2006; Smith & Vardi, 1995). As background light levels increase towards the photopic range, tracer coupling and receptive field sizes of AII cells decrease (Bloomfield & Völgyi, 2004). A reduction in AII–AII cell coupling at higher background light levels is thought to increase the spatial resolution of cone-mediated vision (Demb & Singer, 2012). Our results demonstrating NMDAR modulation of AII cell electrical synapses link NMDARs to the mechanism mediating this light-dependent uncoupling of the AII cell network, which allows the AII amacrine cell to serve different visual functions as lighting conditions change.

AII amacrine cells also share electrical synapses with ON cone bipolar cells. These bidirectional electrical synapses facilitate transmission through the primary rod pathway during scotopic vision and inhibition of OFF cone bipolar and OFF ganglion cells during photopic vision (Münch et al., 2009; Pang et al., 2007; Xin & Bloomfield, 1999). Our tracer coupling experiments suggest that although NMDAR activity reduces coupling between AII cells, coupling is preserved at AII-ON bipolar synapses. Other studies observed a similar pattern of reduced coupling between AII cells and preserved coupling between AII and ON cone bipolar cells in the light-adapted retina (Bloomfield et al., 1997; Petrides & Trexler, 2008). However, because AII cells are coupled to various ON cone bipolar cell subtypes (Petrides & Trexler, 2008), with bipolar cells expressing either connexin 36 or connexin 45 (Deans et al., 2002; Lin et al., 2005), understanding the role of ON cone bipolar cells in NMDAR-mediated plasticity of the AII network remains unclear and warrants further investigation.

NMDARs are commonly associated with plasticity at chemical synapses in the CNS. Here we add to accumulating evidence (Arumugam et al., 2005; Kothmann et al., 2012; Kourosh-Arami et al., 2023; Mathy et al., 2014; Mentis et al., 2002; Pereda & Faber, 1996; Turecek et al., 2014) that NMDARs also contribute to the plasticity of electrical synapses, specifically the electrical synapse coupling AII amacrine cells together. Electrical synapses are an understudied subject in the field of neuroscience. Our study adds to mounting evidence that NMDARs mediate plasticity at both electrical and chemical synapses.

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

## Additional information

### Data availability statement

The data that support the findings of this study are available from the corresponding author upon request.

### Competing interests

The authors declare no competing financial interests.

### Author contributions

C.C., S.P.K. and E.A.N. designed the research; C.C. performed the experiments and analysed the data; C.C., S.P.K. and E.A.N. wrote the paper.

### Funding

Funded by National Institutes of Health grants R01-EY-026514, R01-EY-026882, R01 NS126166 and P30-EY-011374 to E.A.N., R21-EY-033932 to S.P.K. and F31-EY-031578 to C.C.

### Acknowledgements

We thank Paulo Kofuji and Alfonso Araque for lending equipment crucial to the completion of this research.

### Keywords

D-serine, gap junctional conductance, glycine, NMDA receptors, retinal amacrine cells

## Supporting information

Additional supporting information can be found online in the Supporting Information section at the end of the HTML view of the article. Supporting information files available:

**Peer Review History**

