## [Peer Review History · The Journal of Physiology]

Junctional Conductance of Retinal All Amacrine Cell Electrical Synapses is Decreased by NMDA Receptors

Chloe Cable, Sidney P Kuo, and Eric A Newman
DOI: 10.1113/JP286537

Corresponding author(s): Eric Newman (ean@umn.edu)

Review Timeline:

Submission Date:	11-Mar-2024
Editorial Decision:	26-Apr-2024
Revision Received:	22-Oct-2025
Editorial Decision:	03-Dec-2025
Revision Received:	07-Dec-2025
Accepted:	12-Dec-2025

Senior Editor: Nathan Schoppa

Reviewing Editor: Conny Kopp-Scheinflug

Transaction Report:

Dear Dr Newman,

Re: JP-RP-2024-286537 "Junctional Conductance of Retinal All Amacrine Cell Electrical Synapses is Decreased by NMDA Receptors" by Chloe Cable, Sidney P Kuo, and Eric A Newman

Thank you for submitting your manuscript to The Journal of Physiology. It has been assessed by a Reviewing Editor and by 2 expert referees and we are pleased to tell you that it is potentially acceptable for publication following satisfactory major revision.

REVISION CHECKLIST:

We look forward to receiving your revised submission.

Yours sincerely,

David Wyllie
Senior Editor
The Journal of Physiology

REQUIRED ITEMS

- Author photo and profile. First or joint first authors are asked to provide a short biography (no more than 100 words for one author or 150 words in total for joint first authors) and a portrait photograph. These should be uploaded and clearly labelled together in a Word document with the revised version of the manuscript. See Information for Authors for further details.
- You must start the Methods section with a paragraph headed Ethical Approval. A detailed explanation of journal policy and regulations on animal experimentation is given in Principles and standards for reporting animal experiments in The Journal of Physiology and Experimental Physiology by David Grundy *J Physiol*, 593: 2547-2549. doi:10.1113/JP270818). A checklist outlining these requirements and detailing the information that must be provided in the paper can be found at: <https://physoc.onlinelibrary.wiley.com/hub/animal-experiments>. Authors should confirm in their Methods section that their experiments were carried out according to the guidelines laid down by their institution's animal welfare committee, and conform to the principles and regulations as described in the Editorial by Grundy (2015), including an ethics approval reference number. The Methods section must contain a statement about access to food, water and housing, details of the anaesthetic regime: anaesthetic used, dose and route of administration, and method of killing the experimental animals.
- Your manuscript must include a complete Additional Information section, including competing interests; funding; author contributions and acknowledgements.
- Please upload separate high-quality figure files via the submission form.
- Please ensure that the Article File you upload is a Word file.
- Please include an Abstract Figure file, as well as the Figure Legend text within the main article file. The Abstract Figure is a piece of artwork designed to give readers an immediate understanding of the research and should summarise the main conclusions. If possible, the image should be easily 'readable' from left to right or top to bottom. It should show the physiological relevance of the manuscript so readers can assess the importance and content of its findings. Abstract Figures should not merely recapitulate other figures in the manuscript. Please try to keep the diagram as simple as possible and without superfluous information that may distract from the main conclusion(s). Abstract Figures must be provided by authors no later than the revised manuscript stage and should be uploaded as a separate file during online submission labelled as File Type 'Abstract Figure'. Please also ensure that you include the figure legend in the main article file. All Abstract Figures should be created using BioRender. Authors should use The Journal's premium BioRender account to export high-resolution images. Details on how to use and access the premium account are included as part of this email.

Reviewing Editor's Comments:

Your manuscript has been reviewed by two independent reviewers. Both acknowledge the fact that you uncover contradictory findings compared to another paper in the field. I agree with reviewer 2 to repeat that paper's fig 1 in your mice, and then have a better basis to discuss the reason for the differences (other than just possible species differences).

There are quite a number of flaws and lack in rigor as pointed out by reviewer 1. Please take care of addressing these carefully. There is also any information on animal origin, housing and procedures missing. Is there a license number under which you performed your experiments?

Comments to ensure the paper complies with the Statistics Policy:

Please ensure you report precise p values in a revised manuscript - this is the policy of The Journal of Physiology

Senior Editor's Comments:

Thank you for submitting this work for consideration for publication in The Journal of Physiology. As you will read two expert referees have raised some concerns about the work which I feel you would be able to address with further experiments. There needs to be a better reconciliation with why your findings differ from those previously reported and simply stating that it could be due to a species difference. It is not adequate given the difference in experimental approaches used. This needs to be addressed in a revised manuscript.

Referee #1:

Cable et al. investigated the modulatory effect of NMDA signaling on AII-AII amacrine cell coupling in the mouse retina. Using the dual whole-cell recordings, they measured the coupling conductance and examined the NMDA signaling effects using pharmacological dissection. It requires advanced technology, and the authors carefully evaluated the results, shown in Figures 1 and 2. They found that NMDA application reduced the conductance, supported by the reduced coupling conductance by the Mg-free medium and the depolarized holding potential. Then, they examined the effect of D-serine, a co-factor of NMDA receptors. They used three ways to reduce the D-serine level in the retinal tissue; however, those did not affect the reduction of NMDA coupling.

Although using a challenging recording, the authors conducted recordings from many sets of cells and were well summarized. However, the experimental design was not well constructed to address issues. First, the NMDA results are opposite from the results of Kothmann et al. (2013); however, the authors did not address the different results. They conducted Mg-free aCSF and 0mV recording to support the NMDA reduction, which might reduce the conductance for different reasons. Combining with an NMDA agonist or antagonist may verify these data to support NMDA results. Second, light conditions in their experiments are not clearly described. Light or dark levels change dopamine and other chemical

releases, affecting coupling conductance.

Furthermore, as the authors noted, endogenous NMDA existed and reduced the coupling conductance in the free-Mg solution and 0mV recordings. However, the APV application did not change the conductance, but the NMDA bath application reduced the conductance. To fully show the NMDA effect on the coupling, the authors could record the coupling in the Mg-free solution or at 0mV. Lastly, NMDA might reduce the coupling via non-AII cells; a secondary effect might induce the data. To examine this concern, NMDA application might be conducted in the presence of other glutamate, inhibitory, and Na channel blockers.

The originality of the research is unclear. Although data are not consistent, NMDA affecting the AII-AII coupling has been shown by others, such as Kothmann et al. (2013). Co-factors of NMDA receptors are already known.

The article did not fully pursue the physiological mechanisms of NMDA receptor regulations of the coupling. Therefore, the impact on the research is not high at this current manuscript.

Referee #2:

Summary:

All amacrine cells are extensively electrically coupled to other AII cells and to cone bipolar cells. Previous tracing coupling work has found that the strength of AII electrical coupling appears to exhibit plasticity, with NMDA receptor activation possibly being involved. Here the authors performed paired patch recordings from AII cells in slice, and tested the role of various aspects of NMDAR related signaling on gap junction conductance. This is nice work that shows that NMDA receptor signaling decreases gap junction conductance, and is intriguing that it directly contradicts the effect of NMDA receptor activation from a previous retinal study.

Major comments:

- This paper is particularly interesting in that it directly contradicts the finding by Kothmann et al. (2012) about the effect of NMDA receptor activation on AII coupling. However, the discussion point that it could simply be a 'species difference' seems unnecessarily hand-wavy. The authors should repeat the tracer coupling experiments in mice (essentially repeat Fig 1 from the Kothmann paper). Whatever the result, it will be very informative to the issue at hand and provide a much more satisfying conclusion to the current contradiction.

Minor comments:

- The change reported in Figure 3C,D needs to be quantified
- It might be nice to put Figs 3C and F on the exact same y-axis range
- The stats in Fig 4J on the effect of NMDA+d-serine look compelling, and the effect of NMDA+d-serine looks similar to what is shown in 4F, but the example of the 'effect' in 4I does not look compelling or anything like the example shown in 4E. Can the authors comment on this? Also, in 4J it really looks like if you were to add a couple more n then there would be an effect of Ro alone. Commenting on this would be helpful.
- Since the authors do not provide compelling a priori evidence that there is actually a decrease in d-serine in the IP3R2 knockout mice (IP3R2KO), and their results are inconsistent with this idea, it is unclear what these results show. I would suggest possibly removing this
- I understand that this paper is specifically about AII-AII coupling, but some comments on AII-BC coupling, and what is known/unknown about plasticity there could be helpful in the discussion for additional context, as this makes this situation a bit more complex. For example, in the introduction, it's stated that "In bright light, electrical synapses between AII cells uncouple, preventing attenuation of the cone mediated signals through leakage into the AII cell network (Bloomfield & Völgyi,

2004)." However, other results should be kept in mind, such as Munch et al, 2009 (Nat Neuro), found that AII-BC coupling seems fairly robust in light conditions, with signal flow opposite from that in dark conditions, helping establish looming-specific responses.

END OF COMMENTS

We thank the reviewers for their helpful and insightful comments, which have helped to improve the paper significantly. Whenever possible, we have revised the manuscript as suggested by the reviewers. We have made the following major revisions to the manuscript:

- As suggested by the reviewers, we have performed extensive additional experiments, characterizing the effect of NMDAR activation on All amacrine cell coupling using the Neurobiotin tracer technique. The results confirm our junctional conductance results, indicating that NMDA reduces coupling.
- The results of our Neurobiotin tracer experiments are illustrated in a new Fig. 5.
- The Discussion section has been extensively revised to discuss the discrepancy between our results and those of Kothmann *et al.*
- All figures have been revised to show p values within the figures.

Please note that since our original submission to *The Journal of Physiology*, we were invited to contribute to a special issue to be published by the journal. It was agreed that our original submission would be included as an article in this special issue. The submission deadline for the special issue was such that it gave us extra time to perform an extensive series of additional experiments that had been requested by reviewers. That is why we delayed submitting this revision until now.

Changes to the text in the revised, marked manuscript are highlighted in yellow.

Reviewing Editor: Your manuscript has been reviewed by two independent reviewers. Both acknowledge the fact that you uncover contradictory findings compared to another paper in the field. I agree with reviewer 2 to repeat that paper's fig 1 in your mice, and then have a better basis to discuss the reason for the differences (other than just possible species differences).

There are quite a number of flaws and lack in rigor as pointed out by reviewer 1. Please take care of addressing these carefully. There is also any information on animal origin, housing and procedures missing. Is there a license number under which you performed your experiments?

RESPONSE:

As noted above, we have repeated the Kothmann *et al* Neurobiotin tracer experiment and report the results in the revised manuscript and in Fig. 5.

Our responses to Reviewer 1's comments are detailed below.

The information on animal origin, housing, and license number has been added to the Materials and Methods section.

Reviewing Editor: Comments to ensure the paper complies with the Statistics Policy: Please ensure you report precise p values in a revised manuscript - this is the policy of The Journal of Physiology

RESPONSE: We have modified the manuscript to include all p values within the graphs.

Senior Editor: Thank you for submitting this work for consideration for publication in The Journal of Physiology. As you will read two expert referees have raised some concerns about the work which I feel you would be able to address with further experiments. There needs to be a better reconciliation with why your findings differ from those previously reported and simply

stating that it could be due to a species difference. It is not adequate given the difference in experimental approaches used. This needs to be addressed in a revised manuscript.

RESPONSE: We have conducted extensive additional experiments to address the reviewers' concerns about the discrepancy between our results and Kothmann *et al.* (2013). The issue is addressed in detail in the revised Discussion section.

Referee #1: Cable *et al.* investigated the modulatory effect of NMDA signaling on All-All amacrine cell coupling in the mouse retina. Using the dual whole-cell recordings, they measured the coupling conductance and examined the NMDA signaling effects using pharmacological dissection. It requires advanced technology, and the authors carefully evaluated the results, shown in Figures 1 and 2. They found that NMDA application reduced the conductance, supported by the reduced coupling conductance by the Mg-free medium and the depolarized holding potential. Then, they examined the effect of d-serine, a co-factor of NMDA receptors. They used three ways to reduce the D-serine level in the retinal tissue; however, those did not affect the reduction of NMDA coupling.

Although using a challenging recording, the authors conducted recordings from many sets of cells and were well summarized. However, the experimental design was not well constructed to address issues. First, the NMDA results are opposite from the results of Kothmann *et al.* (2013); however, the authors did not address the different results.

RESPONSE: We have addressed the discrepancy between our results and those of Kothmann *et al.* in our revised manuscript. Most importantly, we have repeated the experiment of Kothmann *et al.*, assessing coupling of All amacrine cells using the Neurobiotin tracer method. We conducted an extensive series of tracer experiments. The results confirm our junctional conductance experiments, showing that tracer coupling is decreased with NMDAR activation. The results are shown in the new figure 5.

In addition, we discuss this discrepancy in detail in our revised Discussion section. We state:

“Several factors could account for the discrepancy between our results and those of Kothmann *et al.* Kothmann *et al.* used an NMDAR antagonist in their experiments while we evaluated the effect of both NMDAR agonists and antagonists. Although both studies were done under light-adapted conditions, our experiments were performed in retinal slices using patch pipettes while Kothmann *et al.* used retinal whole mounts and sharp electrodes. We used retinal slices and patch pipettes so that we could directly compare our junctional conductance and tracer coupling results.

Notably, our experiments were performed in mouse while Kothmann *et al.* utilized rabbit. Previous studies have revealed differences in coupling between retinal neurons in different species. While Tsukamoto *et al.* asserted that the canonical scotopic circuitry among mammals is conserved, they also found that some scotopic microcircuitry varied across species (Tsukamoto *et al.*, 2001). While electrical coupling between AIs and ON cone bipolar cells remains conserved between mouse and rabbit, rabbit bipolar cells have significantly more gap junctional contacts compared to mouse. In addition, rabbit bipolar cells are extensively coupled to other bipolar cells of the same class, a pattern that is not observed in the mouse (Sigulinsky *et al.*, 2020). Potential species differences in our study versus that of Kothmann *et al.* may be clarified as studies continue to probe this topic.”

Referee #1: They conducted Mg-free aCSF and 0mV recording to support the NMDA reduction, which might reduce the conductance for different reasons. Combining with an NMDA agonist or antagonist may verify these data to support NMDA results.

RESPONSE: The referee is indeed correct in stating that the changes in conductance resulting from Mg-free aCSF or 0mV may be due to other factors besides NMDA receptor activation. We have revised the wording in the Results section of the manuscript to suggest that NMDARs may play a role in reduction of conductance under these conditions.

Referee #1: Second, light conditions in their experiments are not clearly described. Light or dark levels change dopamine and other chemical releases, affecting coupling conductance.

RESPONSE: We have revised the Methods section to address light levels during experiments. We state:

“Anesthesia, dissection, and slice preparation were performed under normal room illumination. During dual whole-cell patch-clamp electrophysiology and Neurobiotin tracer coupling experiments, lights were dimmed moderately and retinal slices were considered light adapted (Veruki & Hartveit, 2002).”

Referee #1: Furthermore, as the authors noted, endogenous NMDA existed and reduced the coupling conductance in the free-Mg solution and 0mV recordings. However, the APV application did not change the conductance, but the NMDA bath application reduced the conductance. To fully show the NMDA effect on the coupling, the authors could record the coupling in the Mg-free solution or at 0mV.

RESPONSE: We fully agree with the referee and indeed did characterize the effect of NMDA and APV in Mg-free aCSF (Fig. 4E-J). Please note that all experiments illustrated in Figs. 4E-4J and 5-7 were performed in Mg-Free aCSF. We state in the manuscript:

”From this point forward, all experiments investigating the role of NMDARs in modulating All electrical synapses were performed in Mg²⁺-free aCSF containing the chemical synapse antagonist cocktail while clamping both cells at -60 mV holding potential.”

Referee #1: Lastly, NMDA might reduce the coupling via non-All cells; a secondary effect might induce the data. To examine this concern, NMDA application might be conducted in the presence of other glutamate, inhibitory, and Na channel blockers.

RESPONSE: Again, we fully agree with the referee and, because of the possible influence of non-All cells, we added a cocktail of receptor antagonists to block much of the synaptic input onto All amacrine cells. The antagonists we added blocked AMPA, GABAA, and glycine receptors as well as voltage-gated Na⁺ channels (Fig. 3C-D). All experiments illustrated in Figs. 4-7 were conducted in aCSF containing the antagonist cocktail to reduce the influence of non-All cells in our experiments. We write in the Results section of the manuscript:

“Additionally, we evaluated the effect of blocking chemical synapses on junctional conductance by addition of a cocktail of drugs: 10 μM CNQX to block AMPA receptors, 10 μM gabazine to block GABA_A receptors, 1 μM strychnine to block glycine receptors, and 0.3 μM TTX to block voltage-gated Na⁺ channels (Fig. 3C - F). The drug cocktail greatly reduced chemical synaptic

currents recorded from the All cells, as observed in the example traces showing the synaptic currents recorded from one cell of a pair (Fig. 3C, D), without altering the junctional conductance between the two cells of the pair (Fig. 3E, F). The synaptic currents, quantified by measuring the root mean square of the voltage-clamp currents, were reduced $83.6 \pm 9.1\%$ by the drug cocktail. All remaining experiments reported here were conducted in the presence of this chemical synapse antagonist cocktail in order to isolate the effect of our manipulations to electrical synapses.”

Referee #1: The originality of the research is unclear. Although data are not consistent, NMDA affecting the All-All coupling has been shown by others, such as Kothmann et al. (2013). Co-factors of NMDA receptors are already known.

RESPONSE: Our research is original as it demonstrates, for the first time, All electrical synapse plasticity in response to activation of NMDARs using junctional conductance measurements. Our additional experiments with Neurobiotin tracer coupling confirm our electrophysiological findings, and conflict with the results of Kothmann et al, raising an important issue that must be resolved by future research.

The referee is correct in stating that NMDAR co-factors are already known. However, different NMDAR subtypes and the roles of their co-factors under various physiological conditions continues to be studied. We investigated whether D-serine and glycine may differentially affect junctional conductance between Alls, as it may shed further light on the mechanisms at play that induce and modulate such plasticity.

Referee #1: The article did not fully pursue the physiological mechanisms of NMDA receptor regulations of the coupling. Therefore, the impact on the research is not high at this current manuscript.

RESPONSE: We are uncertain what the referee is referring to by “physiological mechanisms”. We describe a range of physiological mechanisms that contribute to the regulation of junctional conductance between All amacrine cells. In addition to characterizing the effects of NMDAR agonists and antagonists on junctional conductance, we describe the role of cofactors D-serine and glycine in regulating junctional conductance. The referee comments that these experiments were unnecessary as the role of these cofactors is already known. We respectfully disagree and believe that our results contribute to elucidating the physiological mechanisms responsible for the regulation of All amacrine cell junctional conductance. The scope of any one study is, by its nature, limited. We believe that the work presented in our paper is novel and represents an important contribution to the field.

Referee #2: All amacrine cells are extensively electrically coupled to other Alls and to cone bipolar cells. Previous tracing coupling work has found that the strength of All electrical coupling appears to exhibit plasticity, with NMDA receptor activation possibly being involved. Here the authors performed paired patch recordings from Alls in slice, and tested the role of various aspects of NMDAR related signaling on gap junction conductance. This is nice work that shows that NMDA receptor signaling decreases gap junction conductance, and is intriguing that it directly contradicts the effect of NDMA receptor activation from a previous retinal study.

Major comments:

Referee #2: This paper is particularly interesting in that it directly contradicts the finding by Kothmann et al. (2012) about the effect of NMDA receptor activation on All coupling. However, the discussion point that it could simply be a 'species difference' seems unnecessarily hand-wavy. The authors should repeat the tracer coupling experiments in mice (essentially repeat Fig 1 from the Kothmann paper). Whatever the result, it will be very informative to the issue at hand and provide a much more satisfying conclusion to the current contradiction.

RESPONSE: We have followed the referee's suggestion and have repeated the experiment of Kothmann et al, assessing coupling of All amacrine cells using the Neurobiotin tracer method. We conducted an extensive series of tracer experiments for this revision. The results confirm our junctional conductance experiments, showing that tracer coupling is decreased with NMDAR activation. The results are shown in the new figure 5. The discrepancy in results is discussed in detail in the revised Discussion section (see above).

Minor comments:

Referee #2: The change reported in Figure 3C,D needs to be quantified

RESPONSE: We have quantified the reduction in synaptic currents and reported the result in the Results section. We write:

"The synaptic currents, quantified by measuring the root mean square of the voltage-clamp currents, were reduced $83.6 \pm 9.1\%$ by the drug cocktail."

Referee #2: It might be nice to put Figs 3C and F on the exact same y-axis range

RESPONSE: Figures 3C and F are not illustrating the same measurement and are not meant to be compared. Fig. 3C depicts changes in synaptic current recorded from one cell in a paired recording, Fig. 3F quantifies changes in junctional conductance between pairs of cells. We have clarified this in the revised text.

Referee #2: The stats in Fig 4J on the effect of NMDA+d-serine look compelling, and the effect of NMDA+d-serine looks similar to what is shown in 4F, but the example of the 'effect' in 4I does not look compelling or anything like the example shown in 4E. Can the authors comment on this? Also, in 4J it really looks like if you were to add a couple more n then there would be an effect of Ro alone. Commenting on this would be helpful.

RESPONSE: We agree and have revised Fig. 4, replacing panel 4I with a more representative experiment.

Prior to beginning the project, we conducted a power analysis using a statistical power of 80% with a type I error rate of $\alpha = 0.05$ and previously reported effect sizes. Based on the power analysis, we have sufficient paired recordings in the data depicted in 4J to achieve statistical power.

Referee #2: Since the authors do not provide compelling a priori evidence that there is actually

a decrease in d-serine in the IP3R2 knockout mice (IP3R2KO), and their results are inconsistent with this idea, it is unclear what these results show. I would suggest possibly removing this

RESPONSE: We agree that there is no direct experimental evidence showing that there are decreased D-serine levels in IP3R2KO mice. However, previous work strongly suggests that this is the case. Takata et al (2011) report decreased D-serine release upon stimulation of the barrel cortex in IP3R2KO mice. Others report that astrocytic D-serine release depends on IP3R-mediated calcium signaling (Sherwood et al 2017, Henneberger et al 2010). Additionally, long-term synaptic depression deficits in IP3R2KO hippocampal slices are reportedly rescued with the addition of D-serine (Pinto-Duarte 2019).

Our IP3R2KO experiments are only one of three different approaches that we used to assess the role of D-serine in regulating junctional coupling between AII cells. All three approaches yielded similar results. We believe that the IP3R2KO experiment is a useful contribution to the issue. We modified the manuscript to reflect these nuances. We write in the Results section:

“Barrel cortex stimulation of IP3R2 knockout mice (IP3R2KO) yields decreased D-serine release compared to WT mice, likely due to decreased astrocytic Ca^{2+} levels (Takato et al 2011). Müller cells, the primary glial cells of the retina (Newman & Reichenbach, 1996), also display decreased glial Ca^{2+} signaling in IP3R2 knockout mice (IP3R2KO) (Biesecker *et al.*, 2016), and therefore, are likely to have decreased D-serine release. We hypothesized that, compared to WT retinas, IP3R2KO retinas treated with NMDA alone would result in a smaller junctional conductance decrease, and that only with the addition of exogenous D-serine would NMDA cause the same magnitude of decrease as seen in the WT retina.”

Referee #2: I understand that this paper is specifically about AII-AII coupling, but some comments on AII-BC coupling, and what is known/unknown about plasticity there could be helpful in the discussion for additional context, as this makes this situation a bit more complex. For example, in the introduction, it's stated that "In bright light, electrical synapses between AII cells uncouple, preventing attenuation of the cone mediated signals through leakage into the AII cell network (Bloomfield & Völgyi, 2004)." However, other results should be kept in mind, such as Munch et al, 2009 (Nat Neuro), found that AII-BC coupling seems fairly robust in light conditions, with signal flow opposite from that in dark conditions, helping establish looming-specific responses.

RESPONSE: We agree with the reviewer that AII-BC coupling further complicates the mechanisms behind NMDAR-mediated plasticity of the AII network. We have revised the manuscript to acknowledge this issue in the Discussion section. We write:

“All amacrine cells also share electrical synapses with ON cone bipolar cells. These bidirectional electrical synapses facilitate transmission through the primary rod pathway during scotopic vision and inhibition of OFF cone bipolar and OFF ganglion cells during photopic vision (Xin & Bloomfield, 1999; Pang *et al.*, 2007; Münch *et al.*, 2009). Our tracer coupling experiments suggest that, while NMDAR activity reduces coupling between AII cells, coupling is preserved at AII-ON cone bipolar synapses. Other studies observed a similar pattern of reduced coupling between AII cells and preserved coupling between AII and ON cone bipolar cells in the light-adapted retina (Bloomfield *et al.*, 1997; Petrides & Trexler, 2008). However, because AII cells are coupled to various ON cone bipolar cell subtypes (Petrides & Trexler, 2008), with bipolar cells expressing either connexin 36 or connexin 45 (Deans *et al.*, 2002; Lin *et al.*, 2005),

understanding the role of ON cone bipolar cells in NMDAR-mediated plasticity of the AII network remains unclear and warrants further investigation.”

Dear Dr Newman,

Re: JP-RP-2025-286537R1 "Junctional Conductance of Retinal All Amacrine Cell Electrical Synapses is Decreased by NMDA Receptors" by Chloe Cable, Sidney P Kuo, and Eric A Newman

Thank you for submitting your manuscript to The Journal of Physiology. It has been assessed by a Reviewing Editor and by 1 expert referees and we are pleased to tell you that it is acceptable for publication following satisfactory revision.

REVISION CHECKLIST:

We look forward to receiving your revised submission.

Yours sincerely,

Nathan Schoppa
Senior Editor
The Journal of Physiology

EDITOR COMMENTS

Senior Editor:

Your revised manuscript has been reviewed by one of the original referees and also closely looked at by the Reviewing Editor. We are pleased to inform you that it has been judged to be acceptable for publication, pending one addition. It appears that the Abstract Figure is a photo of two cells that also appears in Fig. 1A, and, as it now stands, the figure is not informative about the main point of the study. It is Journal policy that accepted manuscripts must include an informative Abstract Figure. Some additional wording around our policy is below.

"The Abstract Figure is a piece of artwork designed to give readers an immediate understanding of the research and should summarize the main conclusions. If possible, the image should be easily 'readable' from left to right or top to bottom. It should show the physiological relevance of the manuscript so readers can assess the importance and content of its findings. Abstract Figures should not merely recapitulate other figures in the manuscript."

Reviewing Editor:

Thank you for your careful revisions. Your manuscript has been evaluated again and no further revisions are suggested.

REFEREE COMMENTS

Referee #1:

This revised article is substantially improved from the previous manuscript. The authors thoroughly addressed my previous concerns.

Now, this reviewer feels that this is a high-impact study in the field, based on detailed measurements and analysis of the dataset using cutting-edge technology. The originality is high. The conclusion is well-supported by their results and discussion.

END OF COMMENTS

Response to Referee

As requested by the Senior Editor, we are submitting a new Abstract Figure that summarizes the main conclusions of our paper in schematic form, drawn using BioRender. A new Abstract Figure Legend is included in the revised text of our paper.

Dear Dr Newman,

Re: JP-RP-2025-286537R2 "Junctional Conductance of Retinal All Amacrine Cell Electrical Synapses is Decreased by NMDA Receptors" by Chloe Cable, Sidney P Kuo, and Eric A Newman

We are pleased to tell you that your paper has been accepted for publication in The Journal of Physiology.

Yours sincerely,

Nathan Schoppa
Senior Editor
The Journal of Physiology

IMPORTANT POINTS TO NOTE FOLLOWING ACCEPTANCE OF YOUR PAPER:

- **IMPORTANT NOTICE ABOUT OPEN ACCESS:** To assist authors whose funding agencies mandate immediate public access to published research findings, The Journal of Physiology allows authors to pay an Open Access (OA) fee to have their papers made freely available immediately on publication.

- You can help your research get the attention it deserves! Check out Wiley's free Promotion Guide for best-practice recommendations for promoting your work at: www.wileyauthors.com/eoo/guide. You can learn more about Wiley Editing Services which offers professional video, design, and writing services to create shareable video abstracts, infographics, conference posters, lay summaries, and research news stories for your research at: www.wileyauthors.com/eoo/promotion.

- If you would like to receive our 'Research Roundup', a monthly newsletter highlighting the cutting-edge research published in The Physiological Society's family of journals (The Journal of Physiology, Experimental Physiology, Physiological Reports, The Journal of Nutritional Physiology and The Journal of Precision Medicine: Health and Disease), please click this link, fill in your name and email address and select 'Research Roundup': <https://www.physoc.org/journals-and-media/membernews>

EDITOR COMMENTS

Senior Editor:

Thank you for the latest revision of your manuscript. It is now acceptable for publication. We appreciate your additional effort in providing an informative abstract figure.